# SAK3 Administration Improves Spine Abnormalities and Cognitive Deficits in App^NL-G-F/NL-G-F^ Knock-in Mice by Increasing Proteasome Activity through CaMKII/Rpt6 Signaling

**DOI:** 10.3390/ijms21113833

**Published:** 2020-05-28

**Authors:** Hisanao Izumi, Ichiro Kawahata, Yasuharu Shinoda, Fred J. Helmstetter, Kohji Fukunaga

**Affiliations:** 1Department of Pharmacology, Graduate School of Pharmaceutical Sciences, Tohoku University, Sendai 980-8578, Japan; hisanao0413@gmail.com (H.I.); kawahata@tohoku.ac.jp (I.K.); yshinoda@tohoku.ac.jp (Y.S.); 2Department of Psychology, University of Wisconsin-Milwaukee, Milwaukee, WI 53201, USA; fjh@uwm.edu

**Keywords:** Alzheimer’s disease, proteasome activity, CaMKII/Rpt6 signaling, spine abnormality

## Abstract

Alzheimer’s disease (AD) is the most common form of dementia and is characterized by neuropathological hallmarks consisting of accumulation of extracellular amyloid-β (Aβ) plaques and intracellular neurofibrillary tangles (NFT). Recently, we have identified a new AD therapeutic candidate, ethyl-8′-methyl-2′,4-dioxo-2-(piperidin-1-yl)-2′H-spiro[cyclopentane-1,3′-imidazo [1,2-a] pyridin]-2-ene-3-carboxylate (SAK3), which ameliorates the AD-like pathology in AppNL-F/NL-F knock-in mice. However, the detailed mechanism underlying the therapeutic effects of SAK3 remains unclear. In this study, we found that SAK3 administration improved the reduced proteasome activity through the activation of CaMKII/Rpt6 signaling in AppNL-F/NL-F knock-in (NL-G-F) mice. Moreover, spine abnormalities observed in NL-G-F mice were significantly reversed by SAK3 administration. Along with this, cognitive impairments found in NL-G-F mice were markedly ameliorated by SAK3. In summary, our data suggest that SAK3 administration increases the activity of the proteasome via activation of the CaMKII/Rpt6 signaling pathway, contributing to improvements in spine abnormalities and cognitive deficits in NL-G-F mice. Overall, our findings suggest that SAK3 might be a new attractive drug candidate, representing a new mechanism for the treatment of AD pathology.

## 1. Introduction

Alzheimer’s disease (AD) is the most common form of dementia and is characterized by neuropathological hallmarks consisting of accumulation of extracellular amyloid-β (Aβ) plaques and intracellular neurofibrillary tangles (NFT), followed by neuronal loss [1,2]. Currently, cholinesterase inhibitors and the N-methyl-d-aspartate receptor antagonist memantine are the only available clinical treatment options, but these drugs provide only symptomatic relief and do not affect disease progression [3]. Therefore, several therapeutics targeting other molecules, such as Aβ and tau, are currently under development [4].

T-type calcium channels characterized by fast inactivation and slow deactivation electrophysiological kinetics [5,6] are known to play various roles in physiological and pathological conditions, such as epilepsy, sleep cycle, and neuropathic pain [7,8,9,10,11]. Recently, several reports showed that T-type calcium channels are involved in cognitive function and synaptic plasticity. Gangarossa et al. demonstrated that Cav.3.2 knockout mice show impaired memory function [12]. Recently, it has been reported that treatment with the T-type selective inhibitor Z944 induced impairments in performance in paired associate learning (PAL) in rats [13]. Furthermore, the T-type calcium channel blocker NNC has been reported to inhibit acute long-term potentiation (LTP) in mouse hippocampal slices [14]. Moreover, we previously reported that the T-type calcium enhancer ST101 potentiated LTP in the rat somatosensory cortex [15].

In the case of AD, the expression of Cav3.1 has been demonstrated to be reduced in the brains of patients with AD and in AD model mice, 3 × Tg mice [16]. They also revealed that blockade of the T-type calcium channel increased Aβ production [16]. Additionally, it has been reported that Aβ-induced synaptic deficiency is mediated by inhibition of T-type calcium channel currents [14]. Considering the above, the T-type calcium channel might be a new attractive target for the development of AD therapeutics. Recently, we developed a new T-type calcium channel enhancer ethyl-8′-methyl-2′,4-dioxo-2-(piperidin-1-yl)-2′H-spiro[cyclopentane-1,3′-imidazo[1,2-a]pyridin]-2-ene-3-carboxylate (SAK3) (Figure 1), which potentiates Cav3.1 and Cav3.3 currents and revealed its therapeutic effects against AD pathology in App^NL-F/NL-F^ knock-in (NL-F) mice [17]. However, the detailed mechanism underlying the inhibition of Aβ accumulation remains unclear.

The ubiquitin-proteasome system (UPS) is a major intracellular protein degradation system [18]. Dysfunction of the UPS causes the accumulation and aggregation of misfolded proteins in the cell, leading to impairment of normal cellular functions and even cell death [18,19]. Previously, reduced proteasome activity was reported in the brains of patients with AD [20]. Several studies had shown that Aβ impairs the activity of the proteasome in vitro [21,22] and in vivo [23]. It has also been shown that Aβ is degraded by the proteasome [21,23]. Importantly, several reports have demonstrated that normal proteasome activity is needed for neuronal functions, including synaptic plasticity and long-term memory formation [24,25,26,27,28], which are well known to be impaired in AD. Therefore, drugs that increase the activity of the proteasome might be beneficial in the treatment of AD [29,30]. The phosphorylation of Rpt6—which is one of the components of proteasome 19S subunit—by Ca^2+^/calmodulin-dependent protein kinase II (CaMKII) is required for enhancement of proteasome activity during long-term memory formation and new spine growth [26,31]. However, no reports are investigating the relationship between the reduction in proteasome activity and dysregulation of CaMKII/Rpt6 signaling in the AD mouse brain. Additionally, there are no therapeutics which increase the proteasome activity via CaMKII/Rpt6 signaling in AD therapy.

In this study, we confirmed that oral SAK3 administration rescues the reduced proteasome activity via improvement of the CaMKII/Rpt6 signaling pathway in NL-G-F mice. Furthermore, synaptic abnormalities observed in the hippocampal cornu ammonis 1 (CA1) and the cortex of NL-G-F mice were ameliorated by SAK3 administration. Along with this, the cognitive impairments in NL-G-F mice were significantly reversed by SAK3. Our results show that SAK3 could represent a new therapy for the treatment of AD pathology.

## 2. Results

### 2.1. SAK3 Administration Ameliorates the Reduction in Proteasome Activity in NL-G-F Mice

Aβ-induced proteasome inhibition has been reported previously [22,32]. Therefore, we first investigated whether proteasome activity was decreased in the NL-G-F mouse brain using fluorogenic peptides for the three major types of proteasome activity (Suc-LLVY-AMC for chymotrypsin-like, Bz-VGR-AMC for trypsin-like, and Z-LLE-AMC for caspase-like activity) following the procedure shown in Figure 1. We found that all three types of proteasome proteolytic activities were significantly decreased in the cortex of NL-G-F mice (Suc-LLVY-AMC: 77.3 ± 3.4%, *p* = 0.0199 vs. wild-type (WT) + vehicle; Bz-VGR-AMC: 87.1 ± 2.7%, *p* = 0.0292 vs. WT + vehicle; Z-LLE-AMC: 73.1 ± 3.6%, *p* = 0.0051 vs. WT + vehicle). SAK3 administration markedly improved the reduced proteasome activities (Suc-LLVY-AMC: 102.4 ± 9.5%, *p* = 0.0118 vs. NL-G-F + vehicle; Bz-VGR-AMC: 102.1 ± 4.7%, *p* = 0.0120 vs. NL-G-F + vehicle; Z-LLE-AMC: 94.7 ± 8.0%, *p* = 0.0369 vs. NL-G-F + vehicle; Figure 2A–C).

### 2.2. SAK3 Administration Improves the Reduction in CaMKII-Rpt6 Signaling in NL-G-F Mice

The CaMKII-Rpt6 signaling pathway has been reported to be required for increasing proteasome activity [24,25]. Although protein kinase A (PKA) is capable of phosphorylating Rpt6 at S120, only CaMKII-dependent Rpt6 phosphorylation is important during long-term memory formation and new spine growth [26,31]. Additionally, we previously reported that SAK3 could increase the autophosphorylation of CaMKII [33,34]. Therefore, we evaluated whether SAK3 administration increases the autophosphorylation of CaMKII (T286) and phosphorylation of Rpt6 (S120). We observed no difference in total protein expression levels of CaMKIIα and Rpt6 among groups (Figure 3A,B,D). However, the autophosphorylation of CaMKIIα (T286) and phosphorylation of Rpt6 (S120) were significantly decreased in the hippocampus of NL-G-F mice compared to WT mice (autophosphorylated CaMKIIα: 64.6 ± 3.7%, *p* = 0.0017 vs. WT + vehicle; phosphorylated Rpt6: 73.6 ± 3.4%, *p* = 0.0080 vs. WT + vehicle), which were significantly improved by SAK3 administration (autophosphorylated CaMKIIα: 96.1 ± 6.2%, *p* = 0.0048 vs. NL-G-F + vehicle; phosphorylated Rpt6: 105.1 ± 3.3%, *p* = 0.0018 vs. NL-G-F + vehicle; Figure 3A,C,E).

### 2.3. SAK3 Administration Rescues Dendritic Spine Abnormalities in NL-G-F Mice

Increased proteasome activity via phosphorylation of Rpt6 at S120 by CaMKII contributes to synaptic strength and new spine growth [25,26]. Therefore, we assessed whether SAK3 administration could promote spine formation in the hippocampal CA1 and cortex using the LY injection method. We observed a significant decrease in the number of spines per 10 μm dendrites in both regions of NL-G-F mice compared to WT mice (CA1: 13.1 ± 0.38, *p* < 0.0001 vs. WT + vehicle; Cortex: 8.9 ± 0.29, *p* = 0.0071 vs. WT + vehicle; Figure 4B,C). SAK3 administration significantly reversed the reduction in spine density in NL-G-F mice (CA1: 14.7 ± 0.37, *p* < 0.0033 vs. NL-G-F + vehicle; Cortex: 10.7 ± 0.43, *p* = 0.0006 vs. NL-G-F + vehicle; Figure 4B,C). Spinal length tended to increase in the NL-G-F mouse brain, but there was no significant difference compared to that in WT mice (Figure 4D,E). The spine head width was significantly decreased in both brain regions examined in NL-G-F mice (CA1: 0.40 ± 0.0057, *p* = 0.0002 vs. WT + vehicle; Cortex: 0.44 ± 0.0046, *p* < 0.0001 vs. WT + vehicle), which was markedly restored by SAK3 administration (CA1: 0.42 ± 0.0041 μm, *p* < 0.0319 vs. NL-G-F + vehicle; Cortex: 0.47 ± 0.0059, *p* < 0.0001 vs. NL-G-F + vehicle; Figure 4F,G). As shown in Figure 4H,I, NL-G-F mice showed an increased proportion of immature spines (thin) and decreased proportion of mature spines (mushroom and stubby) in both brain regions (CA1 thin: 78.7 ± 1.2%, *p* = 0.0004 vs. WT + vehicle; CA1 mushroom: 13.8 ± 0.90%, *p* = 0.0272 vs. WT+ vehicle; CA1 stubby: 7.4 ± 0.74%, *p* = 0.0027 vs. WT + vehicle; Cortex thin: 68.0 ± 1.1%, *p* < 0.0001 vs. WT vehicle; Cortex mushroom: 22.9 ± 0.87%, *p* < 0.0001 vs. WT + vehicle; Cortex stubby: 3.0 ± 0.47%, *p* < 0.0001 vs. WT + vehicle). SAK3 administration significantly decreased the number of immature spines and increased the number of mature spines in NL-G-F mice (CA1 thin: 70.0 ± 1.2%, *p* = 0.0002 vs. NL-G-F + vehicle; CA1 mushroom: 18.8 ± 1.2%, *p* = 0.0052 vs. NL-G-F + vehicle; CA1 stubby: 11.2 ± 0.81%, *p* = 0.0098 vs. NL-G-F + vehicle; Cortex thin: 58.9 ± 1.6%, *p* < 0.0001 vs. NL-G-F vehicle; Cortex mushroom: 30.1 ± 1.4%, *p* < 0.0001 vs. NL-G-F + vehicle; Cortex stubby: 6.3 ± 0.69%, *p* < 0.0014 vs. NL-G-F + vehicle). Since the spine density was decreased in NL-G-F mice, we assessed the expression of synaptophysin and PSD95, markers of pre- and post-synapses, in the hippocampus. Synaptophysin protein levels did not differ among the groups (Figure 5A,B), while the levels of PSD95 were significantly decreased in NL-G-F mice (70.4 ± 4.4%, *p* = 0.0254 vs. WT + vehicle; Figure 5A,C). SAK3 administration markedly restored the reduced expression of PSD95 (101.5 ± 9.2%, *p* = 0.0194 vs. NL-G-F + vehicle; Figure 5A,C).

### 2.4. SAK3 Administration Improves the Memory Impairments in NL-G-F Mice

To investigate whether chronic SAK3 administration could reverse the cognitive deficits in NL-G-F mice, we performed behavioral analyses. In the Y-maze task, the decreased percentage of alternation behaviors observed in NL-G-F mice (54.9 ± 3.2%, *p* = 0.0063 vs. WT + vehicle) were improved by chronic SAK3 administration (70.2 ± 3.6%, *p* = 0.015 vs. NL-G-F + vehicle) without a change in the number of total arm entries (Figure 6A,B). In the novel object recognition task, no groups showed differences in the discrimination index using the same object in the training session (Figure 6C). In the test session, the discrimination index for a novel object in NL-G-F mice was significantly decreased relative to WT mice (−0.074 ± 0.057, *p* = 0.0055 vs. WT + vehicle; Figure 6D). SAK3 administration significantly improved the reduced discrimination index (0.17 ± 0.063, *p* = 0.0351 vs. NL-G-F + vehicle; Figure 6D). In the step-through passive avoidance task, we observed no significant differences in the latency to enter a dark compartment in the absence of a foot shock among the groups (Figure 6E). However, in the test session, the latency to enter a dark compartment was significantly decreased in NL-G-F mice compared to WT mice (108.9 ± 32.7 s, *p* = 0.0027 vs. WT + vehicle; Figure 6F). The reduced latency in NL-G-F mice was markedly increased by SAK3 administration (240.9 ± 28.4 s, *p* = 0.0186 vs. NL-G-F + vehicle; Figure 6F).

### 2.5. SAK3 Administration Improves Performance on the dPAL Task in NL-G-F Mice

To confirm the cognitive improvement effect of SAK3, we conducted the dPAL task, which requires sophisticated processing of information for the proper association of images with specific locations. Prior to being subjected to the dPAL task, mice were trained over several training sessions (initial touch, must touch, must initiate, punish incorrect) to learn the association between touching the stimulus presented on the screen and getting the reward. In all training sessions, we observed no significant differences among the groups (Figure 7B). In the dPAL task, a profound reduction in accuracy was observed in NL-G-F mice, which was significantly rescued by SAK3 administration (two-way repeated ANOVA shows a significant effect of session block: F (9, 162) = 71.04, *p* < 0.0001, a significant effect of mouse type: F (2, 18) = 4.028, *p* = 0.036, no interaction: F (18, 162) = 1.502, *p* = 0.095; Figure 7C). The significantly increased number of correction trials also reflects the poor performance in the dPAL task of NL-G-F mice. SAK3 administration markedly decreased the number of correction trials in NL-G-F mice (two-way repeated-measures ANOVA shows a significant effect of session block: F (9, 162) = 33.6, *p* < 0.0001, a significant effect of mouse type: F (2, 18) = 10.52, *p* = 0.0009, a significant interaction effect: F (18, 162) = 2.244, *p* = 0.0041; Figure 7D). The response latency did not differ among the groups (two-way repeated-measures ANOVA shows a significant effect of session block: F (9, 162) = 10.61, *p* < 0.0001, no effect of mouse type: F (2, 18) = 0.1895, *p* = 0.829, a significant interaction effect: F (18, 162) = 2.353, *p* = 0.0025; Figure 7E). Moreover, the reward collection latency was not significantly different between the three groups (two-way repeated-measures ANOVA shows a significant effect of session block: F (9, 162) = 3.46, *p* < 0.0006, no effect of mouse type: F (2, 18) = 0.2385, *p* = 0.7903, a significant interaction effect: F (18, 162) = 1.955, *p* = 0.015; Figure 7F).

## 3. Discussion

In this study, we report that chronic SAK3 administration improves the reduction in proteasome activity in the brain of NL-G-F mice through improvement in the activity of the CaMKII-Rpt6 signaling pathway. We also revealed that NL-G-F mice show synaptic abnormalities in the hippocampal CA1 and cortex, and SAK3 administration can ameliorate these abnormalities, resulting in the improvement of cognitive impairments.

The UPS is a principal intracellular protein degradation system [18]. UPS dysfunction is associated with neurodegeneration in several protein misfolding diseases, such as AD [35,36], Parkinson’s disease [37,38], amyotrophic lateral sclerosis [39,40], and Huntington’s disease [41]. Accumulation of ubiquitinylated protein has been observed in AD brains [42,43]. A later study showed that the chymotrypsin- and PGPH-like proteasome activities were reduced in the hippocampus, the superior and middle temporal gyri, and the inferior parietal lobule in AD brains without any change in the amount of the proteasomal α and β subunits [20]. Several in vitro studies have demonstrated that proteasome activities are impaired by Aβ [22,32], especially by the A11 antibody-positive Aβ oligomer [44]. Furthermore, Tseng et al. showed a reduction in proteasome activities in a mouse model of AD, 3 × Tg mice [23]. Consistent with these results, we successfully demonstrated that proteasome activities were reduced in brain homogenates from NL-G-F mice (Figure 2). Indeed, Aβ inhibits proteasome activity via direct binding to the 20S proteasome and allosterically impairs the substrate gate in the 20S core particle [44]. The 26S proteasome consists of a 20S core particle and a 19S regulatory particle. Phosphorylation of Rpt6, one of the components of the 19S subunit, regulates the assembly of the 26S proteasome [45]. Moreover, CaMKII-induced Rpt6 phosphorylation at S120 increases the activity of the proteasome [24,25]. In addition to CaMKII, PKA also increases the activity of the proteasome through Rpt6 phosphorylation at S120 [46]. Taken together, we need to define both 20S and 26S proteasome activities in the future study.

Importantly, Jarome et al. showed that CaMKII but not PKA regulated Rpt6 (S120) phosphorylation and enhanced proteasome activity in the amygdala during the formation of long-term memories [31]. Additionally, CaMKII but not PKA mediated phosphorylation of Rpt6 at S120 is required for new spine growth [26]. In this study, we observed a significant decrease in the level of autophosphorylated CaMKII (T286) and phosphorylated Rpt6 (S120) in NL-G-F mice, and SAK3 administration could reverse this effect (Figure 3). Therefore, we conclude that SAK3 administration increases the activity of the 26S proteasome by enhancing the CaMKII-Rpt6 signaling pathway. Remarkably, it has been reported that a reduction in Rpt6 protein expression and proteasome activity was observed in the middle frontal, inferior parietal, and anterior cingulate cortex of patients with AD [47]. Therefore, improvement of Rpt6-regulated proteasome activity by SAK3 administration may represent a new therapeutic target for AD.

The proteasome also contributes to Aβ degradation. In an in vitro study, the proteasome inhibitors lactacystin and epoxomicin significantly decreased the degradation of Aβ [21,23]. In an in vivo study using 3 × Tg mice, injection of epoxomicin into the intracerebral ventricle markedly increased intraneuronal Aβ immunofluorescence in CA1 neurons without changing the levels of amyloid precursor protein (APP) and its relevant C-terminal fragments, suggesting that the proteasome directly degrades Aβ in mouse brain [23]. The metabolism of APP is regulated by the UPS [48]. We previously reported that chronic SAK3 administration could decrease Aβ production in NL-F mice [17] and APP23 mice [49]. In addition, the SAK3 administration upregulated serum- and glucocorticoid-induced protein kinase 1 (SGK1) in NL-F mice [17]. SGK1 mediated the Aβ metabolism [50,51] and the overexpression of SGK1 in the hippocampus recues Aβ pathology in APP/PS1 mice [52]. We assumed that both SGK1 upregulation and increase in proteasome activity would account for the reduction in Aβ plaque size induced by SAK3 administration. However, it is unclear whether SAK3-induced proteasome activation reduced Aβ deposition directly or indirectly via the regulation of APP metabolism.

In AD, synaptic failure and dysfunction are key features associated with memory deficits [53]. Aβ treatment decreases dendritic spine density in neurons in culture [54,55] and acute brain slices [56]. Moreover, several AD mouse models exhibit dendritic spine loss [56,57,58]. Interestingly, oligomeric Aβ-induced spine loss results from the inhibition of new spine formation, but not spine elimination [14]. Moreover, decreased expression of PSD95 has been shown in some AD mouse models [59,60,61]. Notably, it has been reported that hippocampal neurons from NL-F mice show reduced expression of PSD95 and mushroom spines resulting from a reduction in stromal interaction molecule 2 (STIM2) expression, followed by decreased autophosphorylated CaMKII [62]. Here we observed a decrease in the expression of PSD95 (Figure 5) and reduced dendritic spine density (Figure 4) in NL-G-F mice. Morphological analysis of spines in NL-G-F mice revealed an increased number of immature spines and thin spines, and a reduction in the number of mature spines such as mushroom and stubby spines (Figure 4), which are referred to as memory spines [63,64]. Importantly, neuronal functions, such as synaptic plasticity and long-term memory formation, are regulated by the activity of the proteasome [27,28]. Autophosphorylated CaMKIIα acts as a scaffold protein to recruit proteasomes to dendritic spines [65], and phosphorylation of Rpt6 at S120 by CaMKIIα accounts for an increase in proteasome activity and regulation of synaptic strength [24,25]. A phospho-dead mutant variant of Rpt6 (S120A) has a lower activity to accumulate at PSD95-positive synapses in rat hippocampal neurons. However, a phosphomimetic variant of Rpt6 (S120D) increases its delivery to the synapses and is resistant to detergent extraction of PSD95 with Triton X-100 in hippocampal dendrites, suggesting that the phosphorylation of Rpt6 at S120 increases the delivery of proteasomes with spine compartments. In electrophysiological studies, expression of Rpt6 S120A and S120D produce opposing effects on miniature excitatory postsynaptic current (mEPSC) amplitude without any differences in mEPSC frequency, suggesting that changes in proteasome function through phosphorylation of Rpt6 at S120 are involved in synaptic plasticity [25]. Likewise, the proteasome activity mediated by CaMKII-Rpt6 signaling is required for new dendritic spine formation [26]. For example, proteasome inhibitors, mutated Rpt6 (S120A), and the CaMKII inhibitor all decrease the growth of new dendritic spines in hippocampal pyramidal neurons in slice cultures. The bicuculline-induced formation of new spines is also blocked by proteasome inhibitors, an NMDA receptor inhibitor, and mutated Rpt6 (S120A), suggesting that the activity-dependent formation of new spines is mediated by proteasome activity via CaMKII-Rpt6 signaling. Based on these reports, we suggest that SAK3 administration restores synaptic abnormalities through the improvement of proteasome activity regulated by CaMKII-Rpt6 signaling, resulting in amelioration of memory deficits in NL-G-F mice.

In this study, we successfully demonstrated that SAK3 administration could rescue AD pathology, including synaptic deficits and memory impairment, through the improvement of proteasome activity mediated by the CaMKII-Rpt6 signaling pathway. We previously reported that SAK3 potentiates Cav3.1 and Cav3.3 but not Cav3.2 currents [33]. Strikingly, Cav3.1 and Cav3.3, but not Cav3.2, contribute to synaptic development, which is inhibited by oligomeric Aβ via binding to Nogo receptor family 1 (NgR1) followed by activation of Rho kinase (ROCK) signaling [14]. This finding suggests that T-type calcium channel activation is critical for the protection against Aβ-induced synaptotoxicity. We previously reported that SAK3 administration enhanced the release of acetylcholine (ACh) in the hippocampus of mice via T-type calcium channel activation [33]. Moreover, we also found that SAK3 administration promoted glutamate (Glu) release, which was inhibited by treatment with a nicotinic acetylcholine receptor (nAChR) antagonist. Taken together, we hypothesize that SAK3 administration increases ACh release as a result of enhanced T-type calcium channels in the cholinergic terminal, and this promotes glutamatergic transmission via the activation of the nAChR in the glutamatergic terminal. The CaMKII/Rpt6 signaling pathway is enhanced by the stimulation of postsynaptic nAChRs and NMDARs, resulting in proteasome activation through Rpt6 phosphorylation. Thus, increased proteasome activity may account for Aβ degradation and an increase in synaptic strength (Figure 8). In this study, we could not investigate whether tau accumulation is decreased by SAK3 treatment because NL-G-F mice do not reproduce tauopathy, which is one of the pathological hallmarks of AD. Tau has been reported to be a substrate for proteasome and degraded by it [66]. Therefore, in the future, we have to assess the efficacy of SAK3 against tau pathology using other model mice. Taken together, we suggest that SAK3 might represent a new attractive drug candidate with a new mechanism of action for the treatment of AD.

## 4. Materials and Methods

### 4.1. Animals and Experimental Design

NL-G-F mice were generated as described previously [67]. Mice with the same background (C57BL/6J) were used as controls. All animals were kept under conditions of constant temperature (23 ± 1 °C), humidity (55 ± 5%), and a light/dark cycle (light: 9:00–21:00; dark: 21:00–9:00) with free access to standard food and water. All experimental procedures using animals were approved by the Committee on Animal Experiments at Tohoku University.

SAK3 (0.5 mg/kg; structure shown in Figure 1A) was dissolved in distilled water. The experimental schedule is shown in Figure 1B,C. Nine-month-old female mice were orally administered a vehicle or SAK3 by gavage once per day for 3 months. The volume of the solution administered was 0.1 mL per 10 g body weight. The dose of SAK3 was determined from a previous report in which the administered dose rescued the AD pathology in NL-F mice [17]. After a 3-month administration, mice were subjected to behavioral tests—including Y-maze task, novel object recognition task, and step-through passive avoidance task—to assess cognitive function. The day after the behavioral tests were conducted, mice were sacrificed for brain harvesting. Regarding the PAL task, mice were subjected to the task from 9 months of age, and SAK3 was administered from the start to the end of the task.

### 4.2. Proteasome Activity Assay

After the behavioral tasks, cortical brain tissue was dissected and frozen at −80 °C until the analyses were performed. We used 7–8 female mice for each group. Frozen samples were homogenized in ice-cold buffer containing 20 mM Tris-HCl, pH 7.5, 5 mM EDTA, 500 mM NaCl, 5 mM MgCl2, 1% Triton X-100, 1 mM DTT and 2 mM ATP, and centrifuged at 15,000 rpm for 15 min at 4 °C. The protein concentration was determined using a Bradford assay. The same amount of protein was mixed with 200 μM fluorogenic peptides, Suc-LLVY-AMC (Millipore, Bedford, MA, USA) or Bz-VGR-AMC (Enzo Life Science, Farmingdale, NY, USA) or Z-LLE-AMC (Enzo Life Science), and assay buffer (25 mM HEPES, pH 7.5, 0.5 mM EDTA, 0.05% NP-40), followed by incubation for 1 h at 37 °C [68]. After incubation, the fluorescence of the samples was measured at 380/460 nm (Ex/Em) using a fluorometer (FlexStation 3 Multi-Mode Microplate Reader; Molecular Devices, San Jose, CA, USA).

### 4.3. Western Blot Analysis

Western blot analysis was performed as described previously [69]. We used five female mice for each group. Tissues from the dorsal hippocampal region were dissected and frozen at −80 °C until use. Frozen samples were homogenized in ice-cold buffer containing 500 mM NaCl, 50 mM Tris-HCl (pH 7.5), 0.5% Triton X-100, 4 mM EGTA, 10 mM EDTA, 1 mM Na3VO4, 40 mM Na2P2O7·10H2O, 50 mM NaF, 100 nM calyculin A, 50 μg/mL leupeptin, 25 μg/mL pepstatin A, 50 μg/mL trypsin inhibitor, and 1 mM DTT. After homogenization, samples were centrifuged at 15,000 rpm for 10 min at 4 °C. The protein concentration was determined using a Bradford assay, and samples were boiled for 3 min at 100 °C with Laemmli sample buffer (0.38 M Tris-HCl, pH 6.8, 30% 2-mercaptoethanol, 15% glycerol, 12% SDS, and 0.05% bromophenol blue). Equivalent amounts of protein (10 μg) were loaded into SDS-polyacrylamide gels and transferred to Immobilon polyvinylidene difluoride membranes (Merck Millipore Ltd.). After blocking with 5% skimmed milk in TBS-T solution (50 mM Tris-HCl, pH 7.5, 150 mM NaCl, and 0.1% Tween 20) for 30 min at room temperature, membranes were incubated with anti-phospho-CaMKII (1:5000) [70], anti-CaMKII (1:5000) [70], anti-phospho-Rpt6 (1:500; MBS9429032, MyBioSource, San Diego, CA, USA), anti-Rpt6 (1:1000; BML-PW9265, Enzo Life Science), anti-synaptophysin (1:1000; S-5768, Sigma-Aldrich, St Louis, MO, USA), anti-postsynaptic density 95 (PSD95; 1:1000; ab2723, Abcam, Cambridge, UK), and anti-β-actin (1:5,000; A5551, Sigma-Aldrich) for one day at 4 °C. Antibodies were diluted in TBS-T solution. After washing with TBS-T solution, membranes were incubated with the appropriate horseradish peroxidase-conjugated secondary antibodies (1:5000; Southern Biotech, Birmingham, AL, USA) diluted in TBS-T solution for 2 h at room temperature. After several washes, blots were visualized using the Enhanced ChemiLuminescence immunoblotting detection system and the luminescent image analyzer LAS-4000 (Fuji Film, Tokyo, Japan). Protein expression levels were quantified using Image Gauge version 3.41 (Fuji Film).

### 4.4. Dendritic Spine Analysis Using Lucifer Yellow Labeling in Fixed Slices

Intracellular Lucifer yellow (LY) labeling methods were performed as previously described [71]. Following the behavioral tasks, mice were perfused with ice-cold phosphate-buffered saline (PBS, pH 7.4) and then with 4% paraformaldehyde (PFA) under anesthesia. The brain was removed and postfixed in PFA for one day at 4 °C. Coronal sections (250 μm-thick) were prepared using a vibratome (DTK-1000, Dosaka EM Co. Ltd., Kyoto, Japan). Following LY injections into the neurons in the cortex and hippocampal CA1 region, sections were fixed in PFA for one day at 4 °C. After several washes with PBS, slices were incubated in PBS containing 1% bovine serum albumin, 0.3% Triton X-100, and 0.1% NaN3 (blocking solution) for 1 h at room temperature and then reacted with anti-LY antibody (1:1000; A-5750, Invitrogen, Waltham, MA, USA) in blocking solution for 3 days at 4 °C. After PBS washing, sections were incubated with Alexa 488 anti-rabbit IgG (1:500; Invitrogen) for one day at 4 °C. After washing with PBS, sections were mounted on slides with VECTASHIELD (Vector Laboratories, Inc., Burlingame, CA, USA). Immunofluorescent images were analyzed using a confocal laser scanning microscope (TCS SP8, Leica Microsystems, Wetzlar, Germany). We counted 40 dendrites from 4 mice per group. The maximal length and head width of each spine were manually measured using ImageJ software (National Institutes of Health freeware). Spine morphology was assessed according to previously published criteria [72,73]: filopodia spines->2 μm in length, <0.5 μm in width, without a distinct spine head; thin spines-<2 μm in length, <0.5 μm in width, with a neck; stubby spines-<2 μm in length, >0.5 μm in width, <1 length-to-width ratio; mushroom spines-<2 μm in length, >0.5 μm in width.

### 4.5. Behavioral Tasks

#### 4.5.1. Y-Maze Task

Short-term spatial reference memory was assessed using a Y-maze task [17]. We used 7–10 female mice for each group. The apparatus consisted of three identical black Plexiglas arms (50 × 16 × 32 cm). Mice were placed at the end of one arm and allowed to move freely through the maze during an 8 min session. Three consecutive choices of arms were defined as one succeeded alternation. The maximum number of alternations was defined as the total number of arms entered minus two, and the percentage of alternations was calculated as actual alternations/maximum alternations × 100.

#### 4.5.2. Novel Object Recognition Task

The novel object recognition task was performed as described previously [17]. We used 7–10 female mice for each group. In the training session, two objects consisting of a wooden block of the same shape were placed in the test box (35 × 25 × 35 cm^3^), and the mice were allowed to explore for 10 min. Twenty-four hours later, one object was replaced by a novel object, and exploratory behavior was analyzed again for 5 min. After each session, objects were thoroughly cleaned with 70% ethanol to prevent odor recognition. Exploration of an object was defined as rearing on, touching, and sniffing at a distance of <1 cm. A discrimination index was calculated as the ratio of exploratory contacts to familiar and novel objects.

#### 4.5.3. Step-Through Passive Avoidance Task

The step-through passive avoidance task was performed as described previously [17]. We used 7–10 female mice for each group. The test box consisted of light (14 × 10 × 25 cm^3^) and dark (25 × 25 × 25 cm^3^) compartments. The floor was constructed with stainless steel rods, and rods in the dark compartment were connected to an electronic stimulator (Nihon Kohden, Tokyo, Japan). In the fear acquisition session, a mouse was placed in the light compartment and received an electronic shock (0.3 mA, 2 s) when entering the dark compartment. The mouse was removed from the apparatus 30 s later. In the test session, 24 h later, each mouse was placed in the light compartment, and step-through latency was recorded until 300 s elapsed to assess retention level.

### 4.6. PAL Task

#### 4.6.1. Apparatus for the PAL Task

The touchscreen-based apparatus consisted of an operant chamber housed within a sound and light attenuating box. Every trapezoidal-shaped chamber (respective dimensions: big basis = 25 cm; small basis = 6 cm; height = 18 cm) was individually equipped with a food magazine, a house light, a tone generator, a liquid reward dispenser, and a touch screen (Bussey Mouse Touchscreen Chamber, Campden Instruments, U.K.). The food magazine was placed at the small extremity of the trapezoidal chamber, and the touchscreen was located at the opposite end of the chamber. The touchscreen was permanently covered by a black Plexiglas mask with three square windows (side dimensions: length = 7 cm; height = 7 cm) separated by 0.4 cm and located at the height of 3.6 cm from the floor of the chamber. Through these windows, different visual stimuli could be shown on the screen (max. 1 stimulus per window). ABET II Touch software (Campden Instruments) controlled the system and collected data.

#### 4.6.2. Pretraining on the PAL Task

Prior to training, mice were food restricted to maintain approximately 85% of free-feeding body weight. Training of mice for the PAL task was described previously [74,75]. We used 7 female mice for each group. At first, mice were placed in the test chamber with all lights off for 10 min to habituate them to the environment (Habituation 1). The following day, mice were trained to associate a tone with a reward (10% condensed milk). Initially, a tone was played, and a food tray was primed with a reward (150 μL), and 10 s after the mice left the reward tray, a reward was delivered (7 μL) with a tone and illumination of the tray light. This procedure was repeated until a 20 min session ended (Habituation 2). Following habituation, mice were trained to touch a white square stimulus presented pseudo-randomly in one of three windows (initial touch training). After 30 s, the image was removed, and the reward was delivered with a tone accompanied by illumination of the tray light. The touches to the displayed stimulus immediately led to the same outcomes, except the amount of reward (21 μL). Twenty seconds after the mice collected the reward, a new trial was automatically initiated (criterion: completion of 36 trials in 60 min). In the next training (must touch training), the stimulus did not disappear according to the time elapsed, and the mice had to touch the displayed stimulus to receive the reward (criterion: completion of 36 trials in 60 min). In the next training phase (must initiate training), mice were required to nose poke and exit the reward tray before a stimulus was displayed randomly on the screen (criterion: completion of 36 trials in 60 min). Finally, mice were introduced in the last phase of the training required for the PAL task (punish incorrect training). This training was similar to the previous one, but if the mice touched one of the blank screens, the house light was inverted for 5 s, and no reward was given (criterion: 27/36 trials correct in 60 min for 2 consecutive days).

#### 4.6.3. PAL Task

After the successful completion of pretraining, mice were trained on the PAL task (dPAL). At the beginning of the session, each mouse was required to nose poke with exiting the reward tray to begin the first trial. A trial began with the presentation of two novel stimuli on the screen in two of the three locations. One stimulus was the correct S+ and the other was the incorrect S-. There were three different stimuli (flower, plane, and spider) and six possible trial types (Figure 7A). The mice had to touch the correct stimulus to elicit the reward delivery response. In this acquisition session, mice were tested in an unpunished version in which touching the S- was ignored. The criterion for this acquisition phase was the completion of 36 trials in 60 min. After successfully completing the acquisition session, each mouse was tested on a PAL task (dPAL). In the full task, if the mouse’s nose poked the incorrect stimulus, no reward was delivered and the house light was inverted for 5 s, after which correction trials continued until the correct choice was made. A correction trial consisted of the re-presentation of the same stimuli in the same positions as presented in the last trial. Correction trials were excluded from the total count and were not included in the calculation of percent correct. Mice were subjected to a total of 50 daily sessions, with a maximum of 36 trials or 60 min/session.

### 4.7. Statistical Analysis

All data were presented as means ± standard error of the mean (S.E.M). Comparison among multi-groups was evaluated by one-way or two-way analysis of variance (ANOVA) followed by Tukey’s post hoc test. All the statistical analyses were performed using GraphPad Prism 7 (GraphPad Software, Inc., La Jolla, CA, USA). Differences with *p* < 0.05 were considered statistically significant.

## Figures and Tables

**Figure 1 ijms-21-03833-f001:**
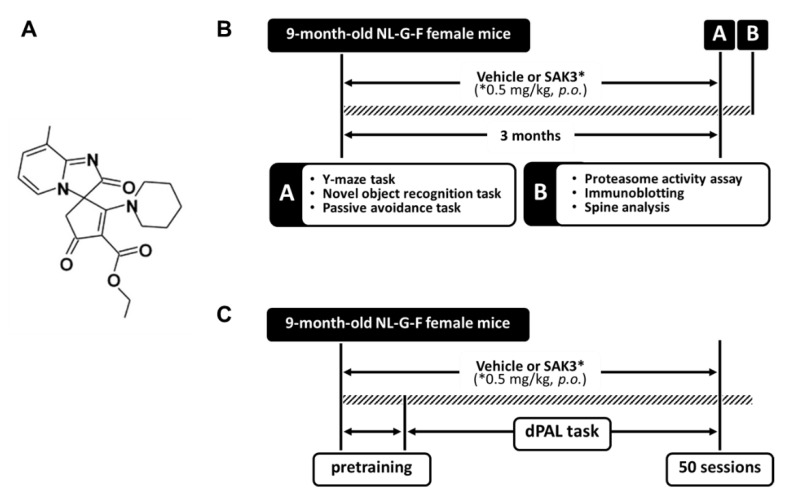
Chemical structure of SAK3 and the experimental schedule. (**A**) Chemical structure of SAK3. (**B**,**C**) Experimental schedule of this study. * Dose of SAK3.

**Figure 2 ijms-21-03833-f002:**
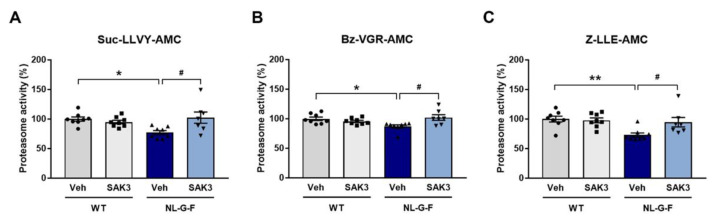
SAK3 administration rescues the decrease in proteasome activity in NL-G-F mice. (**A**) Proteasome activity assay using the fluorogenic peptides Suc-LLVY-AMC (chymotrypsin-like); (**B**) Bz-VGR-AMC (trypsin-like), and (**C**) Z-LLE-AMC (caspase-like) using brain homogenates from the cortical region (*n* = 7–8 per group); Suc-LLVY-AMC: F (3, 27) = 4.814, *p* < 0.01; Bz-VGR-AMC: F (3, 27) = 4.555, *p* < 0.05; Z-LLE-AMC: F (3, 27) = 5.795, *p* < 0.01. Error bars represent standard error of the mean (SEM). * *p* < 0.05, ** *p* < 0.01 vs. vehicle-treated WT mice; # *p* < 0.05 vs. vehicle-treated NL-G-F mice. The symbols (solid dots, squares, triangles and inverted triangle) indicate the individual values.

**Figure 3 ijms-21-03833-f003:**
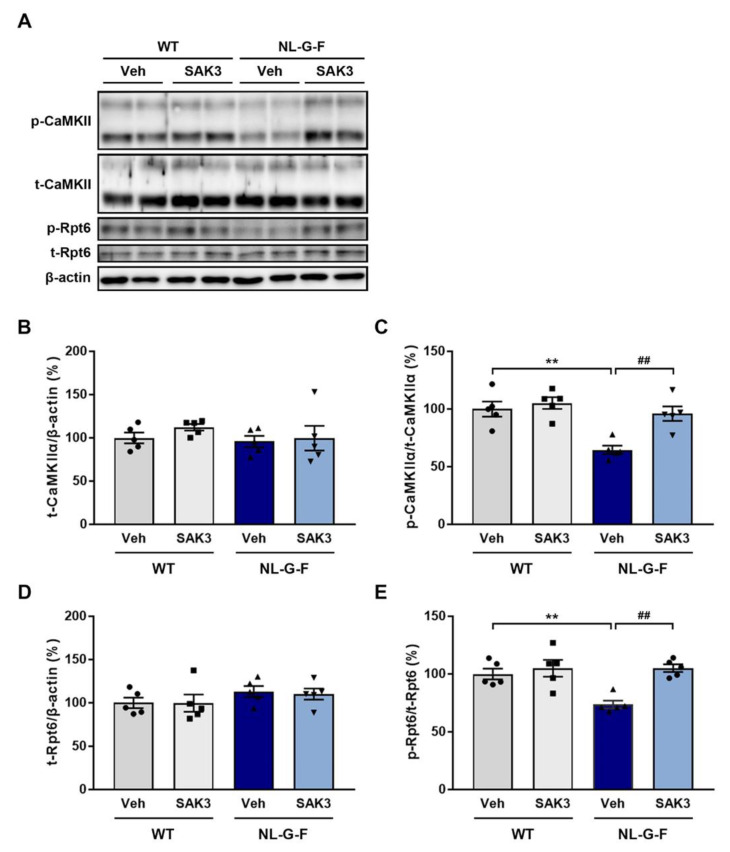
SAK3 administration improves CaMKII-Rpt6 signaling in NL-G-F mice. (**A**) Representative image of western blot membranes containing hippocampal protein probed with antibodies against autophosphorylated CaMKII (T286), CaMKII, phosphorylated Rpt6 (S120), Rpt6, and β-actin. (**B**) Quantitative analyses of CaMKIIα, (**C**) autophosphorylated CaMKIIα (T286), (**D**) Rpt6, and (**E**) phosphorylated Rpt6 (S129) protein levels (*n* = 5 per group); autophosphorylated CaMKIIα: F (3, 16) = 11.01, *p* < 0.01; phosphorylated Rpt6: F (3, 16) = 9.316, *p* < 0.01. Error bars represent SEM. ** *p* < 0.01 vs. vehicle-treated WT mice; ## *p* < 0.01 vs. vehicle-treated NL-G-F mice. The symbols (solid dots, squares, triangles and inverted triangle) indicate the individual values.

**Figure 4 ijms-21-03833-f004:**
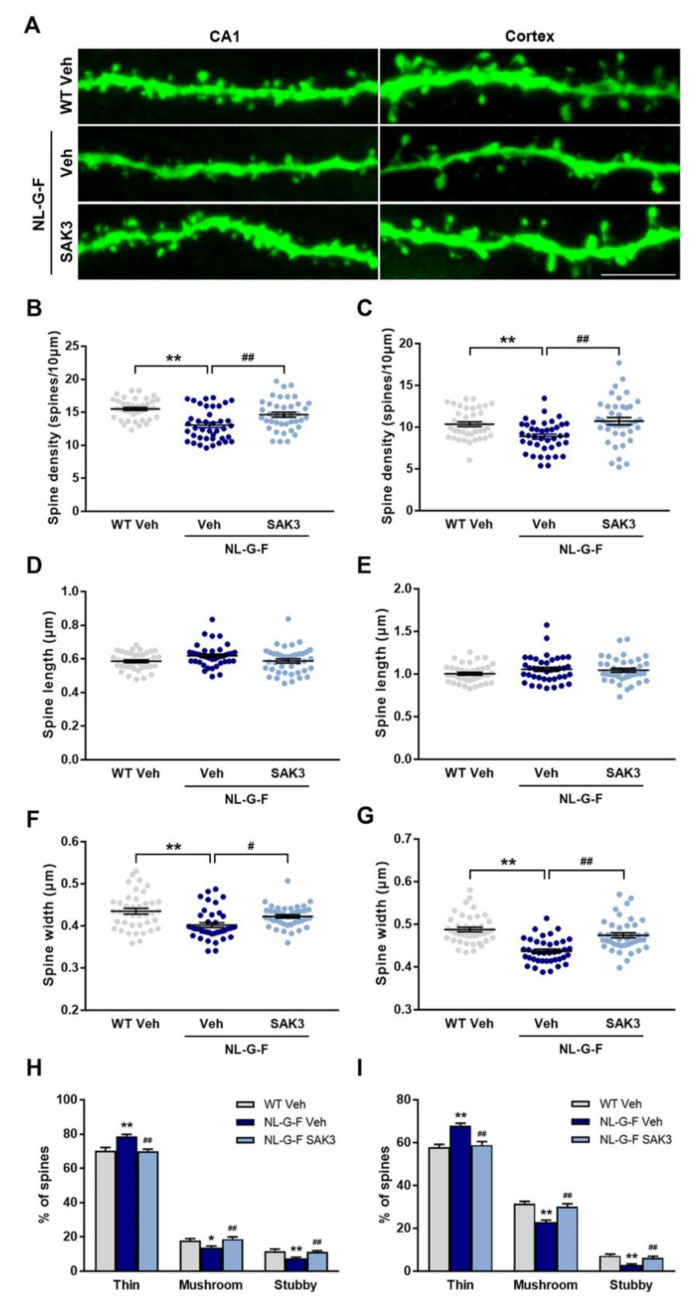
SAK3 administration ameliorates the dendritic spine abnormalities in NL-G-F mice. (**A**) Representative images of dendrites in the hippocampal CA1 and cortex (Scale bar, 5 µm). (**B**) Spine density per 10 µm dendritic length in the hippocampal CA1 and (**C**) cortex (*n* = 40 dendrites from 4 mice per group); CA1: F (2, 117) = 13.67, *p* < 0.01; Cortex: F (2, 117) = 8.302, *p* < 0.01. Spine length in (**D**) the hippocampal CA1 and (**E**) the cortex. Spine width in (**F**) the hippocampal CA1 and (**G**) the cortex; CA1: F (2, 117) = 8.526, *p* < 0.01; Cortex: F (2, 117) = 24.3, *p* < 0.01. (**H**,**I**) The percentage of the type of spines in the hippocampal CA1 and cortex, respectively; CA1 thin: F (2, 117) = 10.91, *p* < 0.01; CA1 mushroom: F (2, 117) = 5.77, *p* < 0.01; CA1 stubby: F (2, 117) = 6.843, *p* < 0.01; Cortex thin: F (2, 117) = 16.93, *p* < 0.01; Cortex mushroom: F (2, 117) = 18.14, *p* < 0.01: Cortex stubby: F (2, 117) = 11.73, *p* < 0.01. More than 2000 spines in the hippocampal CA1 and more than 1500 spines in the cortex were assessed on 40 dendrites from 9–10 neurons from 4 mice per group. Error bars represent SEM. * *p* < 0.05, ** *p* < 0.01 vs. vehicle-treated WT mice; # *p* < 0.05, ## *p* < 0.01 vs. vehicle-treated NL-G-F mice.

**Figure 5 ijms-21-03833-f005:**
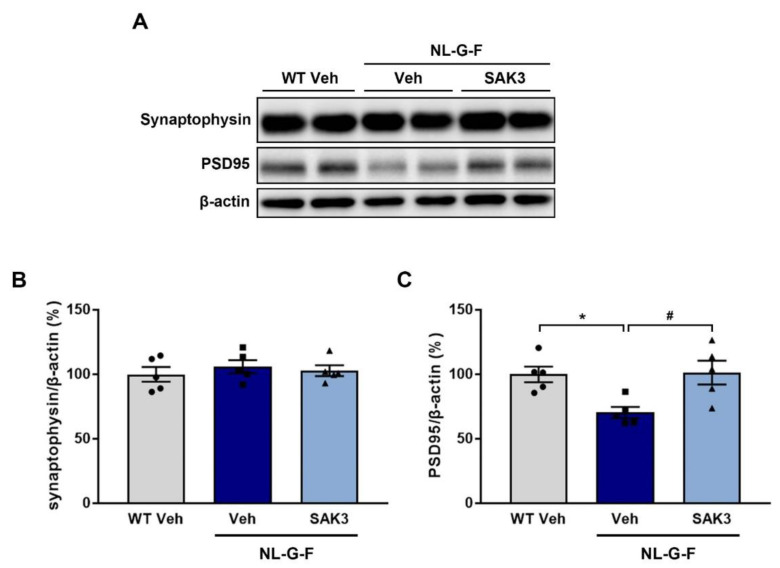
SAK3 administration rescues the reduction in PSD95 expression in NL-G-F mice. (**A**) Representative image of western blot membranes containing hippocampal protein probed with antibodies against synaptophysin, PSD95, and β-actin. Quantitative analyses of (**B**) synaptophysin and (**C**) PSD95 protein levels (*n* = 5 per group). F (2, 12) = 6.52, *p* < 0.05. Error bars represent SEM. * *p* < 0.05 vs. vehicle-treated WT mice; # *p* < 0.05 vs. vehicle-treated NL-G-F mice. The symbols (solid dots, squares and triangles) indicate the individual values.

**Figure 6 ijms-21-03833-f006:**
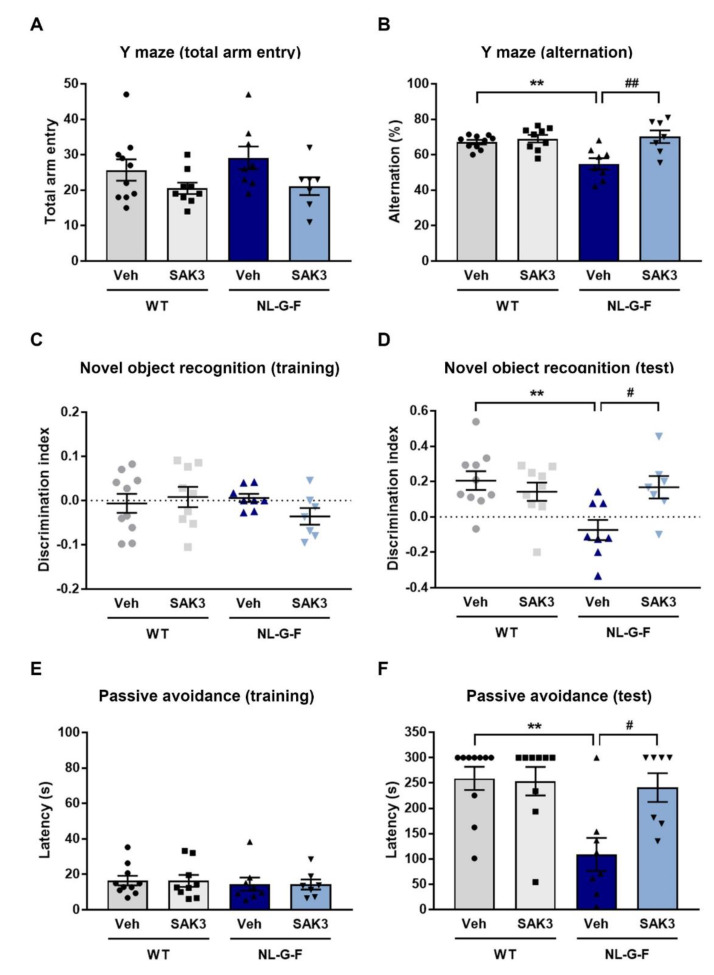
SAK3 administration ameliorates cognitive deficits in NL-G-F mice. (**A**) Number of total arm entries and (**B**) alternations in a Y-maze task (*n* = 7–10 per group). F (3, 30) = 7.678, *p* < 0.01. Discrimination index of object exploration during (**C**) the training session and (**D**) the test session in a novel object recognition task (*n* = 7–10 per group). F (3, 30) = 4.98, *p* < 0.01. Latency to enter the dark compartment in (**E**) the training session and (**F**) the test session of the step-through passive avoidance task (*n* = 7–10 per group). F (3, 30) = 6.425, *p* < 0.01. Error bars represent SEM. ** *p* < 0.01 vs. vehicle-treated WT mice; # *p* < 0.05, ## *p* < 0.01 vs. vehicle-treated NL-G-F mice. The symbols (solid dots, squares, triangles and inverted triangle) indicate the individual values.

**Figure 7 ijms-21-03833-f007:**
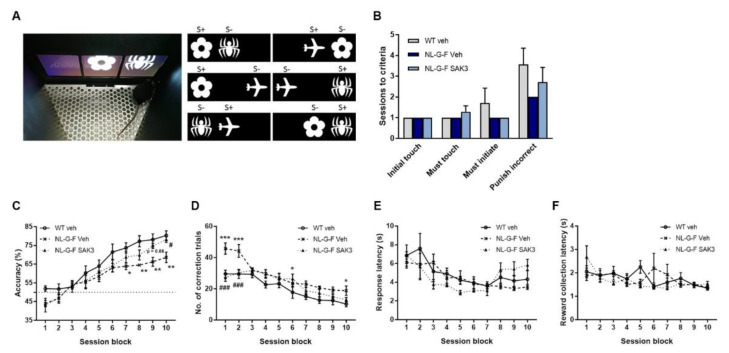
SAK3 administration improves upon the poor performance on the dPAL task in NL-G-F mice. (**A**) Photograph of a mouse performing the dPAL task in the screen operant chamber system and depictions of the six trial types used in the task. (**B**) The number of sessions required to reach the criterion during the pretraining phase (*n* = 7 per group). (**C**) Accuracy, (**D**) number of correction trials, (**E**) response latency, and (**F**) reward collection latency in the dPAL task (*n* = 7 per group). Each session block represents five testing sessions. Error bars represent SEM. * *p* < 0.05, ** *p* < 0.01, *** *p* < 0.001 vs. vehicle-treated WT mice; # *p* < 0.05, ### *p* < 0.001 vs. vehicle-treated NL-G-F mice.

**Figure 8 ijms-21-03833-f008:**
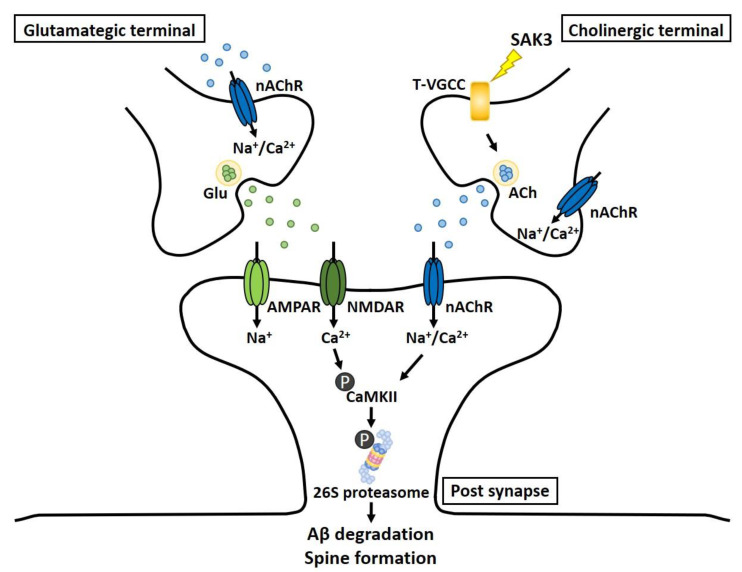
Schematic depicting the mechanism underlying the therapeutic effects of SAK3 administration for AD pathology. SAK3 stimulates T-type calcium channels at cholinergic terminals, triggering intracellular calcium influx, followed by ACh release. Released ACh stimulates nAChRs at glutamatergic terminals, facilitating Glu release. Potentiation of CaMKII/Rpt6 signaling via nAChR and NMDAR activation contributes to an increase in proteasomal activity. Elevated proteasomal activity may account for Aβ degradation and synaptic plasticity.

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
