# Peer review of "SAK3 Administration Improves Spine Abnormalities and Cognitive Deficits in App^NL-G-F/NL-G-F^ Knock-in Mice by Increasing Proteasome Activity through CaMKII/Rpt6 Signaling"

_ijms, 2020, doi:10.3390/ijms21113833_

Round 1
Reviewer 1 Report
It is an interesting manuscript where the authors suggest that SAK3 causes cognitive improvements at the level of synapses mediated through the proteasome. These statements are not clear as they should rule out that:
a) SAK3 has no effect on glial cells. How is microglia inhibited or activated M1 / M2?
b) SAK3 acts on BDNF / Trkb / CREB?
c) Levels of amyloid 42, BACE1 etc ...
d) ROCK / RAC1 cofilin?
e) Mitochondrial status?
Author Response
It is an interesting manuscript where the authors suggest that SAK3 causes cognitive improvements at the level of synapses mediated through the proteasome. These statements are not clear as they should rule out that:
a)SAK3 has no effect on glial cells. How is microglia inhibited or activated M1 / M2?
Ans: This is important question. SAK3 activates T-type channels such as Cav3.1 and Cav3.3. Cav3.1 is predominantly localizes in the post synaptic rejoins of neuronal soma and dendrites and the presynaptic membranes of neuronal terminals in the hippocampus (Front in Neuroanatomy 2016;10:83). We also defined the localization of the soma and neurites of cholinergic neurons but not glial cells in the septum of mouse brain (Neuropharmacology 2017;117:1-13). In addition, SAK3 activates Akt signaling only in the hippocampal pyramidal neurons after oral administration to elicit the neuroprotective action (Neurochem Int 2017;108:272-281). Although we need to examine the secondly effects to glial cells, SAK3 seems to firstly activate T-type channels in neurons.
b) SAK3 acts on BDNF / Trkb / CREB?
Ans: SAK3 elicited its anti-depressive action by the enhancement of neurogenesis in the hippocampus after oral administration in mice (J Pharmacol Sci 2018;137:333-341). In the paper, the enhancements of CREB phosphorylation and BDNF expression were associated with the enhanced neurogenesis in the hippocampus. However, the involvement of BDNF pathway in the proteasome activation by CaMKII remains unclear in the present study.
c) Levels of amyloid 42, BACE1 etc ...
Ans: We have been documented that SAK3 inhibits the enhanced amyloid 42 production in App Knock-in mice (Neuroscience 2018;377:87-97) and APP23 mice (J Pharmcol Sci 2019;139;51-58). “The chronic SAK3 administration could decrease Aβ production in NL-F mice [17] and APP23 mice [49]. In addition, the SAK3 administration upregulated serum- and glucocorticoid-induced protein kinase 1 (SGK1) in NL-F mice [17]. SGK1 mediated the Aβ metabolism [50,51] and the overexpression of SGK1 in the hippocampus recues Aβ pathology in APP/PS1 mice [52]. We assumed that both SGK1 upregulation and increase in proteasome activity would account for the reduction in Aβ plaque size induced by SAK3 administration. However, it is unclear whether SAK3-induced proteasome activation reduced Aβ deposition directly or indirectly via the regulation of APP metabolism.” The descriptions were added in the Discussion.
d) ROCK / RAC1 cofilin?
Ans: This is important to define the mechanism underlying enhanced spine formation. We previously reported that the abnormally increased CaMKII activity elicits the enhancement of GEF activity such a Tiam1 and Kalirin-7, thereby inducing Rac1 activation in the mental retardation model mouse brain (J Neurosci 2011;31:346-358). However, the Rac1 activation by CaMKII elicited increased the filopodia-type spine formation but not mushroom and stubby type spines like present studies. Therefore, the restoration of CaMKII activity did not caused the filopodia-type spine formation in the APP KI mice. However, further studies are required for the involvement of Rac1/cofilin signaling in the APP KI mouse brain.
e) Mitochondrial status?
Ans; This is a hard question to resolve in the present study. We reported that sigma-1 receptor agonist improves the mitochondrial energy production by Ca2+ elevation in mitochondria (Neurochem Int 2019;129:104492) and improves spine formation in the mental retardation mice (Int J Mol Sci 201819:E2811). We need the mitochondrial improvement by SAK3 in the future studies.
Reviewer 2 Report
The work by Izumi and collaborators is an interesting report on the effects of the molecule SAK3 on the proteasome system of a mouse model of amyloid-beta overexpression. The manuscript could be of interest to the readership of the journal. However, the authors should address my following major concerns before the paper is suitable for publication.
- Authors should show how the concentration of amyloid-beta varies in their experimental conditions and correlate this with the activity of the proteasome.
- Authors should specify how many mice they used in their study
- Authors should specify whether the replicates are biological or technical
- Authors need to prove that differences in Fig. 2D (NL-G-F Veh) are significant.
Author Response
Comments and Suggestions for Authors
The work by Izumi and collaborators is an interesting report on the effects of the molecule SAK3 on the proteasome system of a mouse model of amyloid-beta overexpression. The manuscript could be of interest to the readership of the journal. However, the authors should address my following major concerns before the paper is suitable for publication.
- Authors should show how the concentration of amyloid-beta varies in their experimental conditions and correlate this with the activity of the proteasome.
Ans: Although we have not determined the concentration of amyloid-beta in the present study, we have been demonstrated that SAK3 inhibits the enhanced amyloid 42 production in App Knock-in mice (Neuroscience 2018;377:87-97) and APP23 mice (J Pharmcol Sci 2019;139;51-58). In addition, SAK3 enhanced serum-and glucocorticoid-induced protein kinase-1 (SGK-1) gene which is involved in the amyloid beta metabolism. We suggested that both proteasome activation and SGK-1 enhancement by SAK3 are involved in the reduced amyloid plaque formation. The descriptions were added in the Discussion.
- Authors should specify how many mice they used in their study
Ans: We added the description of mouse number used in each study in the Method’s section.
- Authors should specify whether the replicates are biological or technical.
Ans: We used different mice separately with 7-10 female mice in each study such as the behavioral study + proteasome activity, PAL test, immunoblotting and spine morphological study was used. This is enough number to confirm the replication of the studies.
- Authors need to prove that differences in Fig. 2D (NL-G-F Veh) are significant.
Ans: We confirmed the changes in proteasome activity using 7-8 female mouse brain extracts in each group. Then, we confirmed the change in proteasome activity after sucrose density gradient using two mice. We repeated the sucrose gradient procedure twice to confirm that 26S proteasome activity (including Rpt6) was changed without 20S proteasome activity. However, the experimental number is not enough to statistical analyses.
Reviewer 3 Report
In their article the Authors describe the effects of SAK3 administration as therapeutic candidate against AD. The article is well written and interesting and results seem promising for the treatment of the pathology. Data from line 116 to 139 can be inserted in a table because in the text they make difficult following the results. The same suggestion also for the lines 164-176.
Author Response
Comments and Suggestions for Authors
In their article the Authors describe the effects of SAK3 administration as therapeutic candidate against AD. The article is well written and interesting and results seem promising for the treatment of the pathology. Data from line 116 to 139 can be inserted in a table because in the text they make difficult following the results. The same suggestion also for the lines 164-176.
Ans: We understand your comments that the description with each value and its significance. However, the data were represented in the bar graphs in Figs 4 and 7. The P values are also important to confirm the statistics analyses. Therefore, we did not change the current description with bar graphs.
Reviewer 4 Report
I have found the paper written by Izumi et al. very interesting and useful to the IJMS readership. In their work the authors show that SAK3 administration improved the reduced 21 proteasome activity through the activation of CaMKII/Rpt6 signaling in AppNL-F/NL-F knock-in 22 (NL-G-F) mice and I found the experimental design as well as the reported findings sound. Maybe the literature regarding the proteasome could be expanded, including some very recent works regarding molecules used as proteasome activators. Otherwise, in my opinion, the paper could be published as it is
Author Response
Comments and Suggestions for Authors
I have found the paper written by Izumi et al. very interesting and useful to the IJMS readership. In their work the authors show that SAK3 administration improved the reduced 21 proteasome activity through the activation of CaMKII/Rpt6 signaling in AppNL-F/NL-F knock-in 22 (NL-G-F) mice and I found the experimental design as well as the reported findings sound. Maybe the literature regarding the proteasome could be expanded, including some very recent works regarding molecules used as proteasome activators. Otherwise, in my opinion, the paper could be published as it is
Ans: Thank you for your encouragement and suggestion.
Round 2
Reviewer 1 Report
Accept the manuscript
Author Response
Thank you for your comments.
Reviewer 2 Report
I agree with the majority of the authors' response.
- However, the number of mice must be always specified and clear in the different experiments.
- In their response, the authors say "Then, we confirmed the change in proteasome activity after sucrose density gradient using two mice. We repeated the sucrose gradient procedure twice to confirm that 26S proteasome activity (including Rpt6) was changed without 20S proteasome activity. However, the experimental number is not enough to statistical analyses." However, this seems a contradiction to me.
How can the authors state that "In the ATP-regenerating system, 26S proteasome activity (23–26 fractions) was reduced in NL-G-F mouse brain, and this was rescued by SAK3 administration (Fig. 2D). In contrast, in the ATP-depleting system, all groups showed no difference in 20S proteasome activity (Fig. 2E)." if no statistical analysis has been done? Is this trend present in both mice? Are the data averaged values?
Authors need to show that the differences between fraction 23 and 26 are relevant and significant. Data from both mice must be included. Otherwise, their statement is not supported by data and this part should be removed.
Author Response
I agree with the majority of the authors' response.
- However, the number of mice must be always specified and clear in the different experiments.
Ans: We described the number of mice in the text and in the figure legends in each group.
- In their response, the authors say "Then, we confirmed the change in proteasome activity after sucrose density gradient using two mice. We repeated the sucrose gradient procedure twice to confirm that 26S proteasome activity (including Rpt6) was changed without 20S proteasome activity. However, the experimental number is not enough to statistical analyses." However, this seems a contradiction to me.
How can the authors state that "In the ATP-regenerating system, 26S proteasome activity (23–26 fractions) was reduced in NL-G-F mouse brain, and this was rescued by SAK3 administration (Fig. 2D). In contrast, in the ATP-depleting system, all groups showed no difference in 20S proteasome activity (Fig. 2E)." if no statistical analysis has been done? Is this trend present in both mice? Are the data averaged values?
Authors need to show that the differences between fraction 23 and 26 are relevant and significant. Data from both mice must be included. Otherwise, their statement is not supported by data and this part should be removed.
Ans: We used brain extracts from two mice in each group and repeated twice. However the experimental numbers are not enough for the statistical analyses. IN addition, we could not confirm the reduction in the 20S proteasome fraction as previously reported. As the referee recommended, we removed the results of sucrose density gradient and its description from the manuscript.
Thank you for your comment.